# Trajectory-Stabilized Inference for Diffusion-Based Video Inpainting

Zhanhe Zhang[1]   Jiahua Li[1]   Xu Yang[1]   Kun Wei[†1]   Cheng Deng[1]

## Abstract

Video inpainting aims to restore missing regions while preserving spatial and temporal coherence. Diffusion-based methods achieve strong per-frame reconstruction, but their sampling implicitly generates temporally coupled latent trajectories whose long-horizon stability is not explicitly modeled, leading to a trade-off between temporal consistency and structural detail. We revisit video inpainting from the perspective of temporal trajectory stability, viewing temporal inconsistency as instability along time-indexed denoising trajectories rather than an output-level error. Based on this view, we propose an inference-time trajectory stabilization framework that monitors motion-aligned deviation and triggers risk-aware correction only when instability accumulates. It combines sparsely sampled trajectory anchors as stability references with neighborhood-consistent propagation to regulate trajectory evolution while preserving local generative freedom. Implemented as a lightweight control layer in the sampling loop, it selectively contracts unstable trajectories toward motion-consistent manifolds instead of enforcing uniform temporal constraints. Experiments show consistent improvements in temporal coherence and structural fidelity.

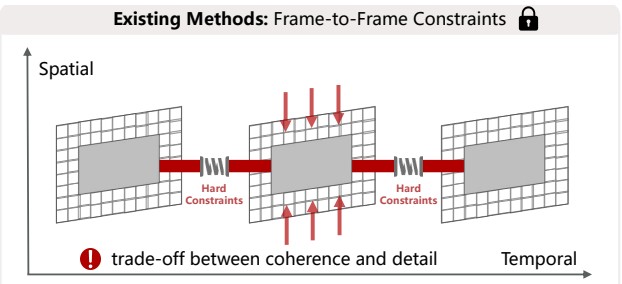

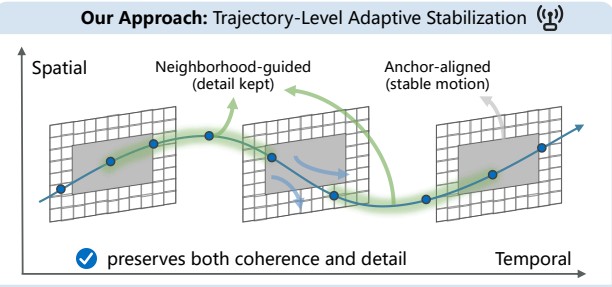

*Figure 1.* Existing methods impose hard frame-to-frame constraints, causing a trade-off between temporal coherence and structural detail. Our method stabilizes time-indexed generative trajectories by applying anchor-aligned corrections only in unstable regions, while stable regions evolve via neighborhood-guided propagation, preserving both coherence and detail.

## 1. Introduction

Video inpainting aims to restore missing regions in a video while preserving spatial and temporal coherence (Xu et al., 2019; Quan et al., 2024). Unlike image inpainting (Pathak et al., 2016; Yu et al., 2018), it is inherently a temporal problem with spatial coupling: the restored content in each frame must remain structurally consistent within the scene, while evolving coherently over time with the dynamics of

neighboring frames. This coupling across time makes video inpainting substantially more challenging than its single-frame counterpart (Wang et al., 2019; Wu et al., 2025).

A widely adopted solution is optical-flow-based propagation (Xu et al., 2019; Li et al., 2022; Zhou et al., 2023; Cho et al., 2025), which transfers pixels or features from neighboring frames into missing regions to maintain cross-frame consistency. While effective in using temporal priors, these approaches remain highly dependent on reliable motion estimation (Li et al., 2022; Zhou et al., 2023). In practice, propagation noise and occlusion variation introduce temporally correlated disturbances that accumulate, eventually causing structural distortion and cross-frame drift in long sequences (Ding et al., 2019; Gowda et al., 2024).

More recently, diffusion-based approaches have been introduced to video inpainting (Zhang et al., 2024; Li et al.,

[1]School of Electronic Engineering, Xidian University, Xi'an 710071, China. Correspondence to: Kun Wei <weikunsk@gmail.com>.

*Proceedings of the 43rd International Conference on Machine Learning*, Seoul, South Korea. PMLR 306, 2026. Copyright 2026 by the author(s).

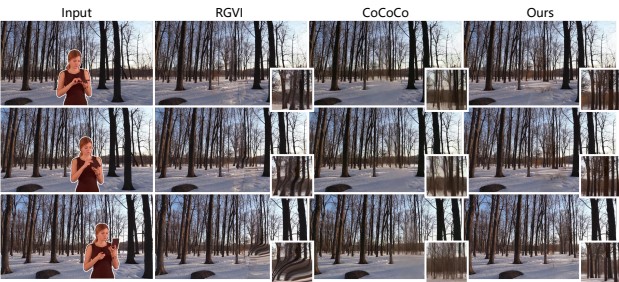

*(a)* Qualitative comparison

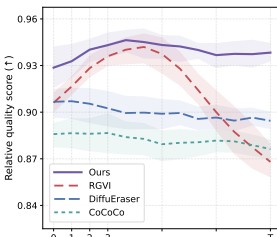 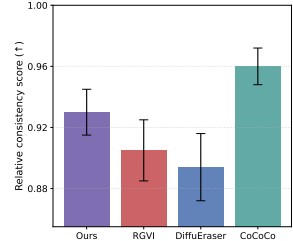

*(b)* Image quality over time.     *(c)* Global temporal consistency.

*Figure 2.* Comparison of temporal stability and perceptual quality. (a) Qualitative results on a challenging long-range sequence. (b) Frame-wise perceptual quality over time, showing that RGVI (Cho et al., 2025) accumulates temporal drift and DiffuEraser (Li et al., 2025) gradually degrades, while our method remains stable. (c) Global temporal consistency, where CoCoCo (Zi et al., 2025) achieves stronger global coherence at the cost of perceptual quality, whereas our method maintains a more balanced trade-off.

2025; Lee et al., 2025b; Xie et al., 2025a). Their step-wise denoising process enables strong semantic completion in regions with weak or missing visual evidence (Green et al., 2024). However, in video settings, temporally coupled latent states must remain compatible with the evolution of neighboring frames during sampling. Existing methods therefore incorporate keyframe alignment or local propagation to regularize temporal consistency during inference (Lee et al., 2025b; Zi et al., 2025; Li et al., 2025). While these strategies help reduce short-range inconsistencies, they operate at the output or step level and cannot reliably detect or correct instability that emerges through persistent deviation along time-indexed trajectories. As a result, stronger temporal regularization tends to restrict generative freedom and suppress fine-scale structure, whereas weaker constraints fail to prevent gradual drift over horizons (Fig. 1).

Building on these observations, we revisit diffusion-based video inpainting from the perspective of trajectory stability. This trade-off is consistently observed in empirical comparisons, where propagation-based and diffusion-based methods exhibit complementary failure modes across long sequences (Fig. 2). We argue that temporal inconsistency is not a per-frame reconstruction error, but a trajectory-level instability along time-indexed generation trajectories in latent space (Daras et al., 2024; Karras et al., 2022). These trajectories are implicitly coupled across frames through motion propagation and cross-frame dependencies, causing deviations to accumulate. Crucially, such instability cannot be reliably detected or corrected by single-step or output-level consistency constraints, as it manifests through persistent deviation along the evolution of the latent trajectory. From this viewpoint, the goal is not to impose stronger output-level supervision, but to regulate trajectory evolution during inference—intervening when instability accumulates, rather than enforcing uniform temporal regularization.

Motivated by this formulation, we develop an inference-time trajectory stabilization framework. The diffusion sampler is augmented with an internal risk state that summarizes motion-aligned deviation along the temporal trajectory and triggers selective stabilization when persistent instability accumulates, avoiding uniform temporal regularization. Corrections are applied in a motion-aligned latent space: unstable regions are softly guided toward anchor-based temporal references, while reliable regions continue to evolve via neighborhood-consistent propagation. As a result, the framework suppresses long-horizon drift where it arises while preserving generative flexibility in stable regions.

Our key contributions are as follows:

- We formulate temporal inconsistency in diffusion-based video inpainting as accumulated instability along motion-aligned denoising trajectories, enabling explicit measurement and inference-time regulation of drift.

- We propose a risk-gated inference-time stabilization mechanism that regulates trajectory evolution to suppress drift without uniformly constraining generation.

- The proposed approach operates at inference time and integrates as a lightweight control layer into frame-wise diffusion pipelines, improving temporal coherence with modest overhead.

## 2. Related Work

### 2.1. Propagation-Based Methods

Propagation-based methods reconstruct missing regions by transferring content from visible frames to occluded ones, typically through optical-flow estimation and feature alignment. Such approaches achieve efficient cross-frame completion and strong short-term consistency by assuming that structural information can be reliably propagated along the temporal axis. Representative examples include E2FGVI (Li et al., 2022), which alleviates propagation discontinuities via end-to-end flow completion; ProPainter (Zhou et al., 2023),

which improves boundary propagation with a dual-domain mechanism; RGVI (Cho et al., 2025), which restores key frames and bidirectionally propagates them to intermediate frames; and HomoGen (Ding et al., 2025), which stabilizes background alignment through homography-based propagation. However, these methods remain susceptible to the accumulation of propagation uncertainty: once local errors occur, they propagate over time and gradually manifest as structural drift (Gowda et al., 2024; Zhu et al., 2025). Rather than improving propagation accuracy itself, our work takes a trajectory-stability perspective, focusing on risk-aware detection and regulation of temporal drift during inference.

## 2.2. Image-Diffusion-Based Methods

With the emergence of diffusion models, many methods combine strong single-frame generative capability with cross-frame constraints to enhance temporal consistency in video inpainting (Croitoru et al., 2023). AVID (Zhang et al., 2024) performs frame-wise diffusion inpainting and enforces local temporal alignment during inference; DiffuEraser (Li et al., 2025) introduces cross-frame guidance into the sampling process to improve stability; CoCoCo (Zi et al., 2025) incorporates motion-aware constraints, while FloED (Gu et al., 2024) and VipDiff (Xie et al., 2025a) integrate optical-flow priors within diffusion sampling to balance generation and temporal coherence. These methods impose local, step-wise cross-frame guidance, yet they typically lack an explicit mechanism to track and intervene on deviation accumulation over long horizons. Consequently, increasing constraint strength often suppresses fine structures, while weaker guidance may allow trajectory drift to accumulate across frames (Daras et al., 2024; Lee et al., 2025a). In contrast, our method operates at the trajectory level in a frame-wise diffusion setting, performing risk-aware, inference-time corrections on the spatio-temporal evolution of the denoising process, enabling stability improvements without over-constraining generation.

## 2.3. Video Diffusion Models

End-to-end video diffusion models jointly model entire video sequences in the spatio-temporal domain, achieving strong generative quality and temporal coherence (Xing et al., 2024). Examples include VideoPainter (Bian et al., 2025), which injects background priors via a context encoder; ROSE (Miao et al., 2025), which models object-removal and side-effect areas; FFF-VDI (Lee et al., 2025b), which shows that a general-purpose model can function as a video completion engine through latent-space propagation; and VidPivot (Xie et al., 2025b), which enhances long-range coherence via anchor-based transitions. While these models provide powerful temporal modeling, they still require expensive training and inference (Wang et al., 2025; Xing et al., 2024). Our work takes a complementary direction:

instead of training a new model, we introduce a lightweight inference-time stabilization mechanism that regulates trajectory evolution using cross-frame priors, improving long-term stability without retraining or modifying the backbone. Our approach is orthogonal to end-to-end video diffusion models and further complements them by providing a drift-aware trajectory stabilization principle (Gong et al., 2019; Li et al., 2018; Deng et al., 2018; Yang et al., 2019; 2020).

## 3. Methodology

We cast diffusion-based video inpainting as the evolution of coupled spatio-temporal denoising trajectories, where temporal inconsistency manifests as drift along these trajectories. Our framework regulates trajectory evolution during inference by detecting motion-aligned deviation and applying selective, anchor-guided stabilization to unstable regions, while reliable areas continue nominal propagation. To preserve long-range coherence, a trajectory-consistent transition prior enforces smooth anchor-to-anchor evolution. Together, these components form a lightweight inference-time stabilization mechanism that improves temporal coherence without modifying the diffusion backbone.

### 3.1. Trajectory-Based Problem Formulation

Let $x_t^i$ denote the VAE latent of frame $i$ at diffusion step $t$. Following standard diffusion models (Ho et al., 2020; Song et al., 2020), the reverse update is

$$x_{t-1}^i = F_\theta(x_t^i) + \varepsilon_t, \tag{1}$$

where $F_\theta$ is the denoising operator and $\varepsilon_t$ is sampling noise.

In a frame-wise video setting, diffusion is applied independently to each frame, but the cross-frame latent set $\{x_t^i\}_{i=1}^T$ at the same diffusion step is coupled through motion, propagation, and cross-frame guidance. We interpret $\{x_t^i\}_{i=1}^T$ as a cross-frame latent trajectory indexed by video time at diffusion step $t$.

As $t$ decreases, the sampler produces a sequence of such trajectories, whose evolution is jointly governed by the denoising operator and cross-frame interactions. Temporal inconsistency arises when these trajectories drift from motion-consistent evolution over long temporal ranges, even if individual frames appear visually plausible.

We introduce an inference-time control term $u_i(t)$ to regulate trajectory evolution:

$$x_{t-1}^i = F_\theta(x_t^i) + u_i(t) + \varepsilon_t, \tag{2}$$

which is activated only when trajectory drift is detected. This formulation casts video inpainting as stabilizing cross-frame latent trajectories during diffusion inference, rather than enforcing uniform per-frame consistency.

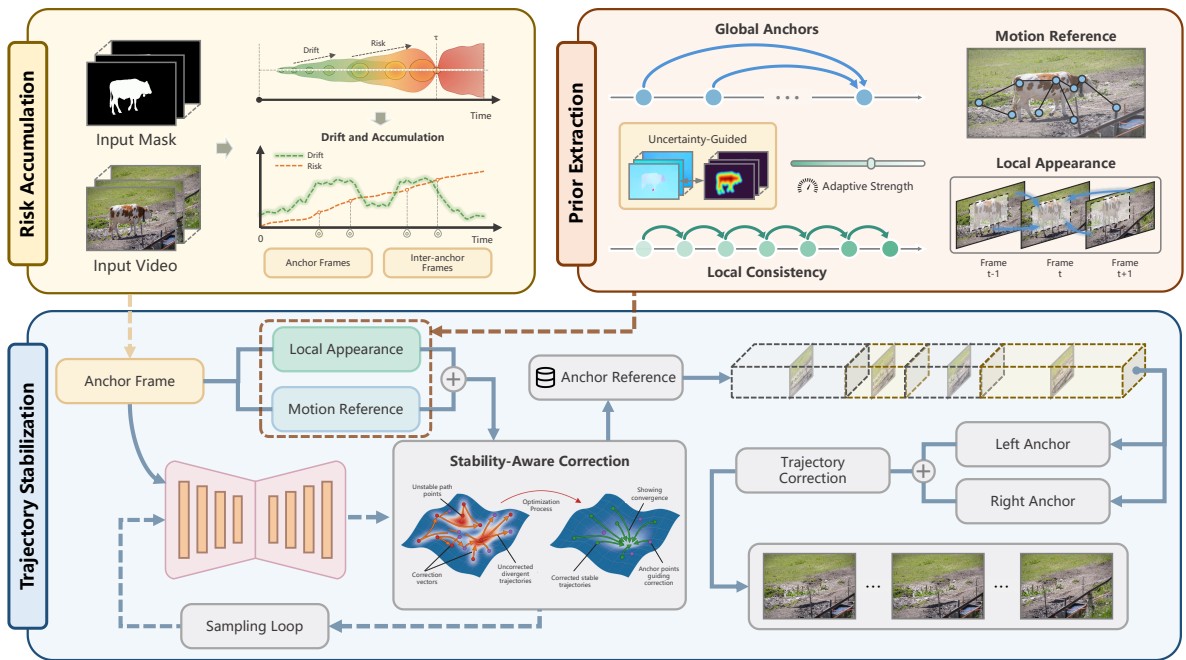

*Figure 3.* Overview of our trajectory-based video inpainting framework. The framework consists of three stages: Risk Accumulation, Prior Extraction, and Trajectory Stabilization. Risk accumulation detects drift and selects anchor frames. Prior extraction builds motion and appearance references. Trajectory stabilization applies anchor-guided correction during diffusion sampling to suppress drift.

## 3.2. Drift-Aware Risk Accumulation

Fig. 3 summarizes the overall inference-time trajectory stabilization pipeline, which we detail below. Given the trajectory formulation in Sec. 3.1, temporal instability corresponds to systematic deviation of the cross-frame latent trajectory from motion-consistent evolution. We therefore measure drift in a motion-aligned latent space. Specifically, given temporal neighbors $N(i)$ and an optical-flow alignment operator $W(\cdot)$ parameterized by $f_{i \leftarrow j}$ on the latent grid (Teed & Deng, 2020; Jaderberg et al., 2015), we define

$$d_i(t) = \sum_{j \in N(i)} w_{ij} \left\| W(x_t^j, f_{i \leftarrow j}) - x_t^i \right\|_1, \quad (3)$$

where $w_{ij}$ is a normalized confidence weight that reflects both the reliability of the motion correspondence and the temporal proximity between frames (Appendix B).

The same neighbors induce a local propagation prior,

$$z_i(t) = \sum_{j \in N(i)} w_{ij} W(x_t^j, f_{i \leftarrow j}), \quad (4)$$

which represents the expected trajectory evolution under locally reliable motion.

We maintain a trajectory risk state for long-horizon drift:

$$R_i(t) = \gamma R_i(t-1) + d_i(t), \quad (5)$$

where $0 < \gamma < 1$. This exponentially weighted accumulation estimates whether deviation persists along the video-

time trajectory rather than being caused by instantaneous noise. When $R_i(t) < \tau$, the system follows nominal motion-guided dynamics; once $R_i(t) \geq \tau$, stabilization is triggered and a controlled update $u_i(t)$ is applied to suppress further drift. We empirically verify that the risk state is predictive of future temporal and perceptual degradation, justifying its use as an inference-time intervention trigger (Appendix C).

## 3.3. Stability-Aware Trajectory Correction

When trajectory drift is detected, we activate a soft, motion-consistent stabilization mechanism that regulates the denoising trajectory rather than overwriting the outcome. The goal is not to force strict agreement across frames, but to gently contract unstable trajectories toward a motion-consistent region while preserving freedom in reliable areas.

We first define a motion-consistent stability geometry in latent space that balances long-range anchor guidance and local motion consistency. Let $\mathcal{A}$ denote a set of structurally reliable anchor trajectories, and let $z_i(t)$ denote a neighborhood prior that provides short-range, motion-aligned propagation when local flow estimates are trustworthy.

A soft confidence mask $\Gamma_i \in [0,1]^{H \times W}$ highlights uncertain or occluded regions, while $(1 - \Gamma_i)$ identifies areas where propagation is reliable. A motion-aligned anchor reference is constructed as

$$q_i^{\text{anc}}(t) = \sum_{a \in \mathcal{A}} \beta_{ia} W(x_t^a, f_{i \leftarrow a}), \quad (6)$$

and the corresponding anchor-induced alignment energy is

$$E_{\text{anc}}(x_t^i) = \frac{1}{2} \left\| \Gamma_i \odot \left( x_t^i - q_i^{\text{anc}}(t) \right) \right\|_2^2, \qquad (7)$$

which provides a stable reference in regions prone to drift due to unreliable local evidence.

To complement it with locally consistent propagation, we introduce a neighborhood alignment term

$$E_{\text{nbr}}(x_t^i, z_i(t)) = \frac{1}{2} \left\| (1 - \Gamma_i) \odot \left( x_t^i - z_i(t) \right) \right\|_2^2, \quad (8)$$

so that anchor guidance dominates uncertain regions, while neighborhood propagation preserves fine-scale temporal continuity where motion estimates are dependable. Together, the two terms define a stability geometry in latent space, where anchors provide long-range temporal reference and neighbors enforce locally motion-consistent evolution. The combined stability potential is

$$E_{\text{stab}}(x_t^i, z_i(t)) = E_{\text{anc}}(x_t^i) + \rho \, E_{\text{nbr}}(x_t^i, z_i(t)). \quad (9)$$

This potential defines a motion-consistent stability basin that favors trajectories aligned with both long-range temporal anchors and locally reliable motion priors. Latent trajectories that deviate from this basin are progressively pulled back toward temporally coherent evolution.

Stabilization is applied through a risk-aware control signal

$$u_i(t) = - \phi(R_i(t)) \, \nabla_{x_t^i} E_{\text{stab}}(x_t^i, z_i(t)), \qquad (10)$$

where $\phi(\cdot)$ is non-negative and monotonically increasing, so that correction strength grows smoothly with accumulated drift risk. The update acts as a trajectory-regularized refinement step: it contracts unstable segments toward the stability basin while avoiding rigid alignment or collapsing diverse generative solutions.

Unlike post-hoc temporal smoothing, this control operates directly on the sampling dynamics, regulating the evolution of denoising trajectories rather than enforcing agreement on completed frames. Intuitively, when the risk state indicates persistent deviation, the control signal suppresses drift amplification along long temporal horizons, whereas stable regions continue to evolve freely under nominal sampling dynamics (see Appendix D for a mechanism interpretation).

### 3.4. Trajectory-Consistent Transition

While stabilization at anchor frames suppresses local drift, long-range temporal coherence depends on how stability propagates along the inter-anchor trajectory path. Rather than enforcing explicit frame-wise agreement, we regulate how the cross-frame latent trajectory evolves between anchors, ensuring a smooth and consistent transition path in trajectory space over time.

Given two anchor trajectories $a_1$ and $a_2$ and an intermediate frame $i$, we construct a motion-aligned, temporally interpolated reference that defines a continuous transition between the two anchor trajectories,

$$\begin{aligned} q_i^{\text{tr}}(t) = \; & \omega_i \, W(x_t^{a_1}, f_{i \leftarrow a_1}) \\ & + (1 - \omega_i) \, W(x_t^{a_2}, f_{i \leftarrow a_2}), \end{aligned} \qquad (11)$$

where $\omega_i$ varies smoothly along the video time axis. This reference defines a continuous trajectory in latent space between anchors, providing a stable evolution path while avoiding abrupt snapping to either endpoint.

Stabilization is restricted to drift-sensitive regions:

$$u_i(t) = - \phi(R_i(t)) \, \Gamma_i \odot \left( x_t^i - q_i^{\text{tr}}(t) \right), \qquad (12)$$

so that high-risk regions are guided toward the anchor-transition trajectory, while low-risk regions follow nominal dynamics and neighborhood propagation.

In this way, the framework does not impose uniform agreement across frames. Instead, it regulates the geometry of inter-anchor latent trajectories, enabling temporally smooth evolution across anchor intervals while preserving fine structural detail and generative flexibility.

## 4. Experiments

This section presents experimental results evaluating the proposed inference-time trajectory stabilization method. We begin with the implementation details, datasets, and evaluation metrics, and then report quantitative and qualitative comparisons against existing approaches. We further conduct ablation studies and performance analyses to examine the contribution and efficiency of each component.

### 4.1. Experimental Settings

**Implementation Details.** All experiments are implemented in PyTorch and run on a single NVIDIA RTX A6000 GPU. We evaluate two frozen diffusion backbones, SDI (Rombach et al., 2022) and BrushNet (Ju et al., 2024), under identical resolution, prompt, optical-flow estimation, and inference schedules to ensure fair comparison. Our trajectory stabilization is inserted as an inference-time control layer, without modifying the diffusion backbone or applying post-hoc temporal blending. Additional implementation details and ablations are provided in Appendix A.

**Dataset.** We evaluate the proposed method on three public video inpainting benchmarks: DAVIS (Perazzi et al., 2016), HQVI (Cho et al., 2025), and VPBench (Bian et al., 2025), covering motion patterns, scene structures, and occlusion types. Following the ProPainter protocol, we evaluate 50 DAVIS sequences with fixed masks for comparison. HQVI

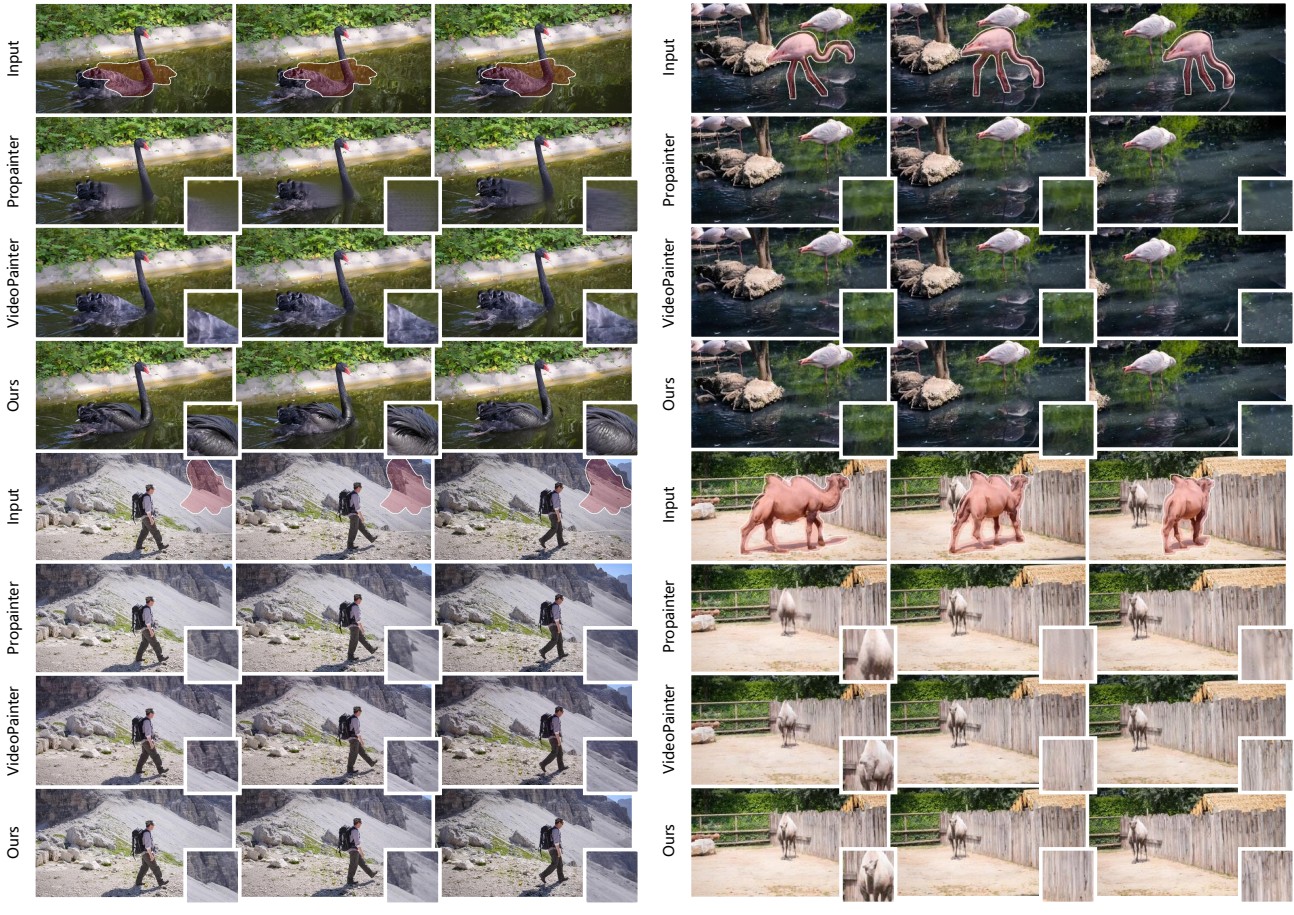

*(a)* Video completion.

*(b)* Object removal.

*Figure 4.* Qualitative comparison on (a) video completion and (b) object removal. For each sequence, we show the input frames and the results of ProPainter, VideoPainter, and our method on challenging scenes with motion and long-range temporal evolution.

provides high-quality test cases with complex motion and occlusion, while VPBench covers scenes and object categories with long-range temporal dependencies. Unless otherwise specified, we use VPBench-L as the primary test set for long-sequence evaluation.

**Metrics.** We follow standard evaluation protocols for video inpainting and report PSNR, SSIM, LPIPS, VFID, and Speed. PSNR and SSIM evaluate frame-level reconstruction fidelity (Wang et al., 2004), while LPIPS measures perceptual similarity in a learned feature space (Zhang et al., 2018). To assess video-level perceptual quality and temporal consistency, we further report VFID, which captures distributional differences between generated and reference videos (Unterthiner et al., 2018). Speed is measured under a unified hardware and inference configuration to evaluate computational efficiency.

### 4.2. Experimental Results

**Quantitative Analysis.** We evaluate all methods on the DAVIS and HQVI benchmarks under a unified inference pro-

tocol. Table 1 reports PSNR, SSIM, LPIPS, VFID, and the average per-frame runtime. Across both datasets, augmenting diffusion-based pipelines with our inference-time trajectory stabilization yields consistent improvements over their original counterparts across different backbones. The gains are most evident in SSIM, LPIPS, and VFID, indicating enhanced structural fidelity and improved long-range temporal consistency under challenging motion and occlusion. On HQVI, which emphasizes long-horizon temporal coherence, the augmented pipelines achieve the best or near-best VFID among diffusion-based approaches. Notably, improvements become more pronounced on long sequences and high-risk segments, highlighting the effectiveness of selective trajectory stabilization under accumulated drift. Overall, the resulting pipelines offer a more favorable trade-off between perceptual quality, temporal stability, and runtime efficiency under identical inference settings.

**Qualitative Analysis.** Figure 4 presents qualitative comparisons on representative video completion and object removal sequences involving large motion, long-range tempo-

*Table 1.* Quantitative comparison on DAVIS and HQVI. All methods are evaluated under identical experimental settings. Bold values indicate the best results. "+Ours" denotes inserting our inference-time trajectory stabilization into the same diffusion backbone. "–" indicates cases where VFID is not applicable under the original pipeline.

| Methods | DAVIS | | | | HQVI | | | | Speed |
|---|---|---|---|---|---|---|---|---|---|
| | PSNR↑ | SSIM↑ | LPIPS↓ | VFID↓ | PSNR↑ | SSIM↑ | LPIPS↓ | VFID↓ | (s/frame) |
| FuseFormer (Liu et al., 2021) | 32.54 | 0.9702 | 0.0371 | 0.138 | 29.92 | 0.9365 | 0.0498 | 0.2727 | 0.114 |
| E2FGVI (Li et al., 2022) | 33.01 | 0.9721 | 0.0328 | 0.116 | 30.63 | 0.9427 | 0.0401 | 0.1885 | 0.085 |
| ProPainter (Zhou et al., 2023) | **34.19** | **0.9759** | **0.0221** | **0.098** | 30.62 | 0.9413 | 0.0388 | 0.2128 | **0.083** |
| RGVI (Cho et al., 2025) | 29.03 | 0.9168 | 0.0691 | 0.198 | 30.66 | 0.9527 | 0.0335 | 0.1825 | 0.453 |
| VideoPainter (Bian et al., 2025) | 33.29 | 0.9681 | 0.0371 | 0.137 | 30.33 | 0.9521 | 0.0374 | 0.2052 | 0.752 |
| SDI (Rombach et al., 2022) | 30.48 | 0.9482 | 0.0546 | – | 29.13 | 0.9425 | 0.0621 | – | 0.965 |
| SDI + Ours | 32.21 | 0.9715 | 0.0339 | 0.128 | **31.85** | **0.9530** | **0.0325** | **0.1795** | 0.797 |
| BrushNet (Ju et al., 2024) | 31.04 | 0.9536 | 0.0478 | – | 29.24 | 0.9436 | 0.0512 | – | 0.892 |
| BrushNet + Ours | 33.70 | 0.9735 | 0.0248 | 0.112 | 31.55 | 0.9524 | 0.0332 | 0.1818 | 0.864 |
| DiffuEraser (Li et al., 2025) | 31.80 | 0.9650 | 0.0450 | 0.160 | 27.26 | 0.8880 | 0.0819 | 0.2962 | 0.921 |
| DiffuEraser + Ours | 32.48 | 0.9692 | 0.0396 | 0.142 | 28.95 | 0.9045 | 0.0708 | 0.2585 | 0.965 |
| CoCoCo (Zi et al., 2025) | 32.51 | 0.9642 | 0.0305 | 0.123 | 29.72 | 0.9488 | 0.0410 | 0.2350 | 0.857 |
| CoCoCo + Ours | 33.04 | 0.9718 | 0.0274 | 0.111 | 31.20 | 0.9516 | 0.0340 | 0.1980 | 0.892 |

*Table 2.* Ablation study of components in our trajectory-stabilized inference framework under the same evaluation setting. AS denotes adaptive anchor selection (uniform sampling when disabled); SA/SN denote anchor-/neighborhood-level stabilization; TB denotes transition blending.

| | Comp. | | | | PSNR↑ | SSIM↑ | LPIPS↓ | VFID↓ |
|---|---|---|---|---|---|---|---|---|
| | AS | SA | SN | TB | | | | |
| (a) | ✗ | ✓ | ✓ | ✓ | 31.72 | 0.9686 | 0.0390 | 0.150 |
| (b) | ✓ | ✗ | ✓ | ✓ | 30.98 | 0.9631 | 0.0468 | 0.191 |
| (c) | ✓ | ✓ | ✗ | ✓ | 31.85 | 0.9692 | 0.0373 | 0.145 |
| (d) | ✓ | ✓ | ✓ | ✗ | 31.93 | 0.9697 | 0.0365 | 0.140 |
| (e) | ✓ | ✓ | ✓ | ✓ | **32.21** | **0.9715** | **0.0339** | **0.128** |

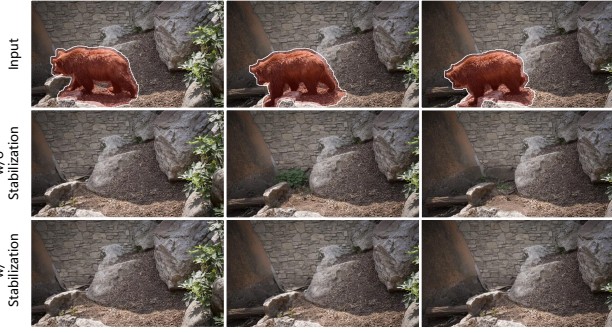

*Figure 5.* Ablation of trajectory stabilization in diffusion-based video inpainting. Without stabilization (middle), cross-frame deviations accumulate along the diffusion trajectory, leading to visible temporal drift. With trajectory stabilization enabled (bottom), such drift is effectively suppressed, resulting in improved temporal coherence across frames.

ral evolution, and complex background structures. Under these challenging conditions, existing propagation-based approaches often exhibit temporal artifacts such as texture flickering, inconsistent object boundaries, and gradual structural drift as temporal distance increases. In contrast, diffusion pipelines augmented with our inference-time trajectory stabilization produce more coherent and visually consistent results across frames: object boundaries remain stable, background appearance evolves smoothly, and fine-scale textures are preserved without introducing noticeable temporal flicker, even under persistent occlusion and viewpoint changes. These observations are consistent with the improvements in LPIPS and VFID reported in Table 1, further supporting the effectiveness of the proposed approach in mitigating long-range temporal drift.

### 4.3. Ablation Study

We perform a one-component-out ablation on DAVIS under an identical inference protocol to isolate the effect of each module in our trajectory-stabilized inference.

As reported in Table 2, removing any component leads to consistent degradation in reconstruction quality and temporal metrics, indicating that the stabilization pipeline relies on complementary mechanisms rather than a single dominant heuristic. In particular, disabling adaptive anchor selection (a) reduces performance across all metrics, suggest-

ing that anchor placement is critical for providing reliable long-range references and preventing misaligned corrections from amplifying drift. This observation is consistent with our anchor density analysis in Appendix E, where selective (risk-aware) anchoring remains consistently stronger than uniform sampling across different anchor budgets.

Removing anchor-level stabilization (b) causes the largest drop, especially in temporal metrics, highlighting that local propagation alone cannot prevent long-horizon deviation once drift starts accumulating. Disabling neighborhood-level stabilization (c) noticeably degrades VFID and LPIPS, reflecting increased local flicker and reduced short-range coherence in regions where motion is reliable. Removing transition blending (d) also harms VFID, indicating that stable inter-anchor evolution benefits from a smooth transition prior rather than abrupt anchor-to-anchor switching. With all modules enabled (e), the full model achieves the best overall results, supporting that anchor references, neighborhood consistency, and transition regularization jointly contribute to stable long-horizon inference.

Figure 5 qualitatively corroborates these trends: without stabilization, drift manifests as progressive flicker and structural inconsistency; removing neighborhood stabilization increases local temporal artifacts; and removing transition blending introduces discontinuities across anchor intervals.

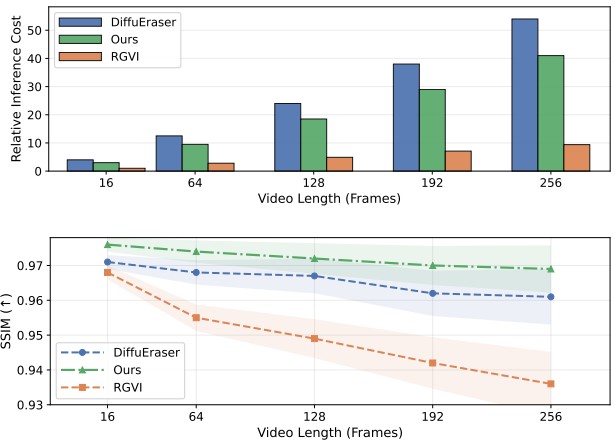

*Figure 6.* Effect of video length on inference cost and temporal stability. The top panel shows the scaling of inference cost as video length increases, while the bottom panel reports SSIM (↑) across different sequence lengths. Our trajectory-stabilized inference maintains stable reconstruction quality as video length grows.

### 4.4. Long-Video Evaluation

To further assess long-horizon robustness, we evaluate all methods on VPBench-L, which contains video sequences with extended temporal spans and complex motion. As shown in Fig. 6, inference cost increases with video length

*Table 3.* Quantitative comparison on the VPBench-L benchmark for long-video evaluation. All methods are evaluated under identical experimental settings. Bold numbers indicate the best results.

| Methods | VPBench-L | | | |
|---|---|---|---|---|
| | PSNR↑ | SSIM↑ | LPIPS↓ | VFID↓ |
| ProPainter | 20.11 | 0.840 | 0.112 | 0.48 |
| VideoPainter | 22.19 | 0.850 | 0.099 | 0.17 |
| SDI | 22.55 | 0.856 | 0.100 | 0.25 |
| SDI + Ours | 22.95 | 0.866 | 0.088 | 0.18 |
| BrushNet | 22.75 | 0.862 | 0.098 | 0.24 |
| BrushNet + Ours | **23.11** | **0.871** | **0.085** | **0.17** |
| DiffuEraser | 22.60 | 0.858 | 0.101 | 0.26 |
| DiffuEraser + Ours | 22.98 | 0.868 | 0.089 | 0.18 |
| CoCoCo | 19.51 | 0.660 | 0.162 | 0.62 |
| CoCoCo + Ours | 20.70 | 0.705 | 0.142 | 0.46 |

for all methods, while propagation-based pipelines exhibit progressive degradation in SSIM as sequence length grows, reflecting accumulated temporal drift under long-range inference and error propagation effects. In contrast, our trajectory-stabilized inference maintains stable SSIM with lower computational growth than frame-wise generation, indicating improved robustness to long-range temporal evolution in practical long-video settings.

Quantitative results on VPBench-L are reported in Table 3. Across different diffusion backbones, augmenting the pipelines with our inference-time stabilization improves PSNR and SSIM while reducing LPIPS and VFID, indicating simultaneous gains in reconstruction fidelity and temporal consistency. Notably, these improvements become more pronounced for longer sequences, suggesting that explicitly regulating denoising trajectories is particularly beneficial when temporal dependencies span extended horizons, where error accumulation and drift are more severe.

## 5. Conclusion

We argue that long-horizon video diffusion failures are better understood as trajectory-level instability rather than output-level inconsistency, and demonstrate that such instability can be selectively regulated at inference time. Based on this formulation, we introduce a trajectory-aware inference framework that selectively regulates denoising trajectories via motion-aligned anchors, without retraining the backbone. This perspective enables targeted stabilization during sampling while preserving generative flexibility. Extensive experiments demonstrate improved temporal coherence and perceptual fidelity with competitive efficiency. Additional analyses are provided in Appendix F.

## Acknowledgments

Our work is supported in part by the National Key R&D Program of China (No. 2023YFC3305600) and the National Natural Science Foundation of China (U25B2048, 62132016, 62406238).

## Impact Statement

This work proposes an inference-time stabilization framework for diffusion-based video inpainting, improving temporal coherence without retraining or modifying the backbone model. The method is lightweight and plug-and-play, which may benefit applications such as film restoration, content editing, and privacy-preserving object removal by enabling more consistent video completion. As with other generative editing techniques, the approach could be misused for misleading or harmful content manipulation; we therefore emphasize responsible use and conduct experiments only on controlled benchmark settings. More broadly, the trajectory-centered perspective explored in this work may inform future research on stability and regulation in generative temporal models.

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

## A. Additional Implementation Details

Our method operates strictly at inference time and does not modify the diffusion backbone or its training procedure. All backbone weights are kept frozen during evaluation. The proposed stabilization is applied as an inference-time control mechanism within diffusion-based video inpainting pipelines that perform frame-wise denoising.

Bidirectional optical flow between adjacent frames is estimated using RAFT (Teed & Deng, 2020), and occluded regions are handled by the recurrent flow completion module from ProPainter. Flow is used only for motion-aligned warping on the latent grid and for mask or region alignment; we do not impose any pixel-level temporal loss or apply post-hoc temporal blending.

Trajectory risk is accumulated online during diffusion sampling with fixed parameters $\gamma = 0.9$ and $\tau = 0.25$ across all datasets and backbones. Given the input video and masks, anchor selection and stabilization are deterministic and introduce no additional supervision or retraining. All diffusion-based methods are evaluated using the same fixed, template-based prompt that provides only coarse scene context. All baselines share identical input resolution, optical-flow estimation, and inference schedules to ensure fair and reproducible comparisons.

## B. Risk Definition and Properties

**Confidence weight.** In the main paper, the confidence weight $w_{ij}$ controls how strongly a temporal neighbor $j$ contributes to both the deviation measurement $d_i(t)$ and the neighborhood propagation prior $z_i(t)$. In our experiments, $w_{ij}$ reflects both motion reliability and temporal proximity.

Given the forward optical flow $f_{i\leftarrow j}$ and the backward flow $f_{j\leftarrow i}$, motion reliability is measured by forward–backward consistency:

$$\alpha_{ij} = \exp\left(-\frac{\|f_{i\leftarrow j} + f_{j\leftarrow i}\|_2^2}{\sigma^2}\right). \tag{13}$$

We further apply an exponential temporal decay

$$\rho_{ij} = \exp(-\lambda|i - j|). \tag{14}$$

The normalized confidence weight is defined as

$$w_{ij} = \frac{\alpha_{ij}\,\rho_{ij}}{\sum_{k\in N(i)} \alpha_{ik}\,\rho_{ik} + \epsilon}, \tag{15}$$

where $\epsilon$ is a small constant for numerical stability. Temporally closer and more motion-consistent neighbors therefore contribute more strongly to both $d_i(t)$ and $z_i(t)$.

**Risk-state recursion.** Recall that the motion-aligned deviation at diffusion step $t$ is

$$d_i(t) = \sum_{j\in N(i)} w_{ij}\left\|W(x_t^j, f_{i\leftarrow j}) - x_t^i\right\|_1, \tag{16}$$

and the risk state is updated as

$$R_i(t) = \gamma R_i(t-1) + d_i(t), \qquad 0 < \gamma < 1. \tag{17}$$

**Closed-form expansion.** Unrolling the recursion yields

$$R_i(t) = \sum_{k=0}^{t-1} \gamma^k\, d_i(t - k), \tag{18}$$

which shows that $R_i(t)$ is an exponentially weighted accumulation of recent motion-aligned deviations. Consequently, persistent deviation leads to a sustained increase of $R_i(t)$, while short-lived mismatches are rapidly downweighted.

**Boundedness.** If the instantaneous deviation is bounded as $0 \leq d_i(t) \leq d_{\max}$, then

$$0 \leq R_i(t) \leq \frac{d_{\max}}{1 - \gamma}, \tag{19}$$

so the risk state remains well-defined and bounded for all $t$. Larger $\gamma$ corresponds to longer temporal memory and increases sensitivity to long-horizon drift.

## C. Risk–Degradation Calibration

This appendix evaluates whether the proposed risk state $R_i(t)$ serves as a meaningful *inference-time trigger* for trajectory stabilization. Because trajectory drift is a latent phenomenon without direct supervision, we validate $R_i(t)$ by examining whether elevated risk states are systematically associated with large *future observable degradation* in the rendered video.

### C.1. Observable Degradation Proxies

Let $\{\hat{x}^i\}$ denote the reconstructed video and $\{x^i\}$ the ground truth. We characterize frame-level observable degradation using two complementary LPIPS-based discrepancies that capture spatial fidelity and motion-consistent temporal quality. The motion-aligned temporal discrepancy is defined as

$$\Delta_{\text{temp}}^i = \text{LPIPS}\left(\hat{x}^i, \ \mathcal{W}(\hat{x}^{i-1}, f_{i \leftarrow i-1})\right), \tag{20}$$

which measures temporal inconsistency after warping the previous frame into the current one using optical flow. The spatial reconstruction discrepancy is

$$\Delta_{\text{spat}}^i = \text{LPIPS}(\hat{x}^i, x^i), \tag{21}$$

which evaluates perceptual distortion relative to the ground-truth frame. We combine these two terms into a single observable degradation proxy

$$D^i = \alpha \, \Delta_{\text{spat}}^i + (1 - \alpha) \, \Delta_{\text{temp}}^i, \qquad \alpha = 0.5, \tag{22}$$

which jointly reflects spatial reconstruction quality and motion-consistent temporal coherence. No tuning is performed for $\alpha$.

### C.2. Empirical Calibration

The risk state $R_i(t)$ is computed online during diffusion sampling from motion-aligned latent deviations and does not depend on ground truth or final outputs, whereas $\Delta_{\text{spat}}^i$ and $\Delta_{\text{temp}}^i$ are evaluated only after rendering and provide an external probe of spatial and temporal quality loss. We therefore analyze how frame-level samples distribute in the joint observable space $(\Delta_{\text{spat}}, \Delta_{\text{temp}})$ as a function of their associated risk values.

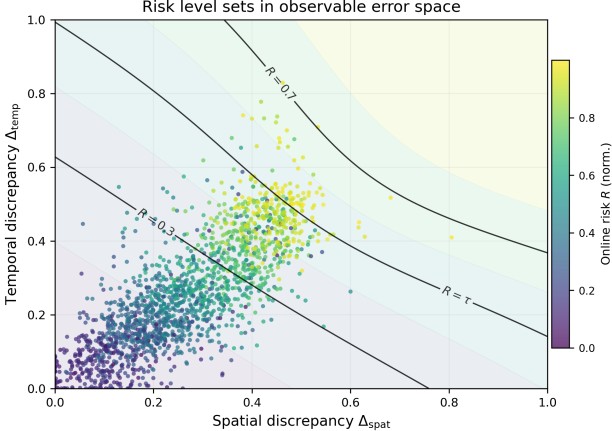

*Figure 7.* Risk level sets in observable error space. Each point is a frame-level sample located by spatial discrepancy $\Delta_{\text{spat}}$ and motion-aligned temporal discrepancy $\Delta_{\text{temp}}$, and colored by the online risk state $R_i(t)$. Black curves denote iso-risk contours. Low-risk samples cluster near the origin, while high-risk samples concentrate in regions of jointly large spatial distortion and temporal inconsistency, indicating that $R_i(t)$ provides a coherent geometric predictor of future observable degradation beyond any single metric.

Figure 7 reveals a clear geometric organization of risk in the observable error space. Low-risk states concentrate in the lower-left region, where both spatial and temporal discrepancies are small, whereas high-risk states progressively occupy the upper-right region characterized by simultaneous perceptual distortion and temporal inconsistency. The iso-risk contours form smooth, monotonic level sets, indicating that $R_i(t)$ aligns with a coherent direction of increasing degradation rather than tracking either $\Delta_{\text{spat}}$ or $\Delta_{\text{temp}}$ alone.

Importantly, this calibration confirms that states deemed risky during sampling are statistically enriched with substantially larger future degradation in the final video. Although $R_i(t)$ does not estimate a ground-truth drift magnitude, it functions as a reliable *early-warning signal* that predicts entry into regions of high observable error, which is precisely the requirement for an effective inference-time stabilization trigger.

## D. Why Risk-Adaptive Stabilization Works

This section explains why the proposed risk-adaptive stabilization improves long-range temporal coherence without collapsing fine structures. The key mechanism is a three-level selectivity that ensures corrections are applied only when, where, and how they are truly needed.

**Temporal selectivity.** Stabilization is activated only when the accumulated risk exceeds a threshold, i.e., when $R_i(t) \geq \tau$. Because $R_i(t)$ integrates motion-aligned deviations over time, short-lived mismatches caused by noise, imperfect flow, or transient occlusions are downweighted, while persistent trajectory drift leads to sustained growth of $R_i(t)$ and eventually triggers intervention. This temporal filtering prevents unnecessary corrections in stable regimes.

**Spatial selectivity.** Stabilization is further modulated by the confidence mask $\Gamma_i \in [0,1]^{H \times W}$. Anchor-guided correction is applied primarily in regions with high uncertainty or occlusion (large $\Gamma_i$), while regions supported by reliable motion evidence (small $\Gamma_i$) continue to evolve through neighborhood-consistent propagation. This spatial gating avoids global alignment and preserves fine-scale textures in well-constrained areas.

**Trajectory-level selectivity.** Importantly, stabilization operates on the intermediate latent states $x_t^i$ during diffusion sampling, rather than on the final reconstructed frames. Thus, the method regulates the *evolution* of denoising trajectories in latent space, contracting unstable segments toward motion-consistent references instead of enforcing post-hoc agreement on completed pixels. This trajectory-level intervention prevents oversmoothing while suppressing long-range drift.

**Energy-descent interpretation.** Recall the stability potential

$$E_{\text{stab}}(x_t^i, z_i(t)) = E_{\text{anc}}(x_t^i) + \rho\, E_{\text{nbr}}(x_t^i, z_i(t)), \tag{23}$$

and the risk-adaptive control

$$u_i(t) = -\phi(R_i(t))\nabla_{x_t^i} E_{\text{stab}}(x_t^i, z_i(t)), \qquad \phi(\cdot) \geq 0. \tag{24}$$

For sufficiently small effective step size, this update corresponds to a descent step on the stability potential:

$$E_{\text{stab}}(x_t^i + u_i(t)) \leq E_{\text{stab}}(x_t^i) - \phi(R_i(t))\big\|\nabla E_{\text{stab}}(x_t^i)\big\|_2^2 + \mathcal{O}\big(\phi(R_i(t))^2\big). \tag{25}$$

Hence, higher accumulated risk induces stronger contraction toward the motion-consistent stability basin, while low-risk regions follow the nominal sampling dynamics with minimal additional regularization. Unlike frame-space temporal smoothing, this risk-gated latent-space descent preserves the model's ability to generate fine structures while selectively suppressing unstable trajectory drift.

## E. Anchor Density Analysis

We analyze the effect of anchor frame density on inference-time trajectory stabilization. All settings are kept identical except for the number of anchor frames used during sampling. Unless otherwise specified, all results are obtained on the DAVIS benchmark under the same inference protocol as in the main experiments.

Figure 8 reports normalized reconstruction quality and temporal consistency, where the best-performing configuration under each metric is scaled to 1.0. As the number of anchor frames increases, both reconstruction performance and temporal

consistency improve in the low-density regime, indicating more stable trajectory evolution. Beyond a moderate anchor budget, further increasing anchor density yields diminishing returns and may slightly reduce performance, suggesting that excessive anchoring can over-constrain the denoising trajectory.

Across all anchor budgets, adaptive anchor selection consistently outperforms uniform sampling. This indicates that effective trajectory stabilization depends more on *where* anchors are placed than on anchor density alone. These observations support our design choice of risk-aware, inference-time anchoring, which selectively allocates stabilization only when trajectory drift emerges, rather than uniformly enforcing temporal constraints.

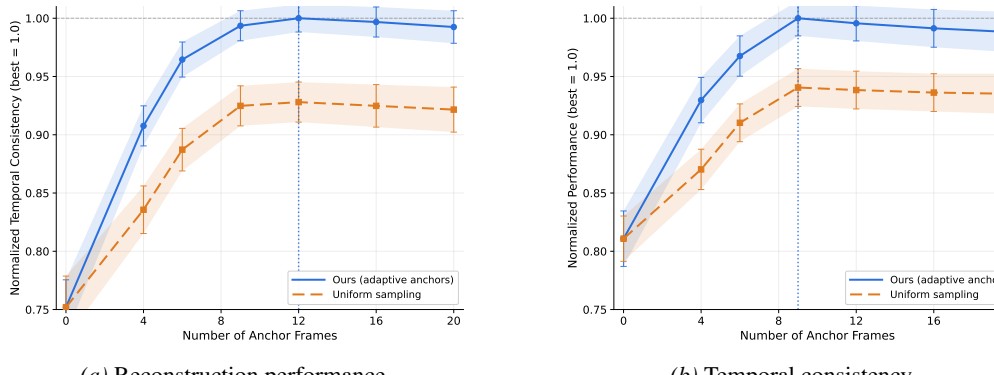

*(a)* Reconstruction performance.  *(b)* Temporal consistency.

*Figure 8.* Effect of anchor frame density on inference-time trajectory stabilization. Metrics are normalized so that the best-performing configuration equals 1.0. Adaptive anchor selection consistently outperforms uniform sampling across different anchor budgets.

## F. Additional Analyses and Limitations

### F.1. Sensitivity to Risk Parameters

We examine the sensitivity of the proposed framework to the two parameters governing risk accumulation and activation: the memory coefficient $\gamma$ and the threshold $\tau$.

The coefficient $\gamma$ controls the temporal horizon over which deviation is accumulated. Smaller values emphasize short-term mismatch, while larger values capture long-horizon drift. We evaluate $\gamma \in \{0.7, 0.8, 0.9, 0.95\}$ and observe stable performance for $\gamma \in [0.85, 0.95]$, indicating robustness to the choice of temporal memory.

The threshold $\tau$ determines when stabilization is triggered. Excessively small values lead to over-regularization, while overly large values delay necessary intervention. Testing $\tau \in \{0.15, 0.25, 0.35, 0.45\}$ shows that $\tau \in [0.2, 0.3]$ consistently yields strong performance across benchmarks, suggesting a broad and stable operating regime.

### F.2. Limitations

Despite improved temporal stability, the framework relies on the availability of informative motion and anchor cues. When optical flow becomes unreliable over large regions (e.g., under extreme camera motion or severe disocclusion), both deviation estimation and anchor construction may be noisy, leading to delayed or less precise stabilization. Similarly, if a region remains occluded for an extended interval without any reliable anchor observation, the transition reference becomes under-constrained, and inference increasingly depends on the generative prior. These limitations stem from the quality of available temporal cues rather than from the trajectory-stabilization mechanism itself.

### F.3. Supplementary Qualitative Results

We include additional qualitative examples to complement the main results. Figures 9 and 10 provide further comparisons on challenging sequences, highlighting the temporal coherence and structural stability of our method under complex motion and long-range propagation.

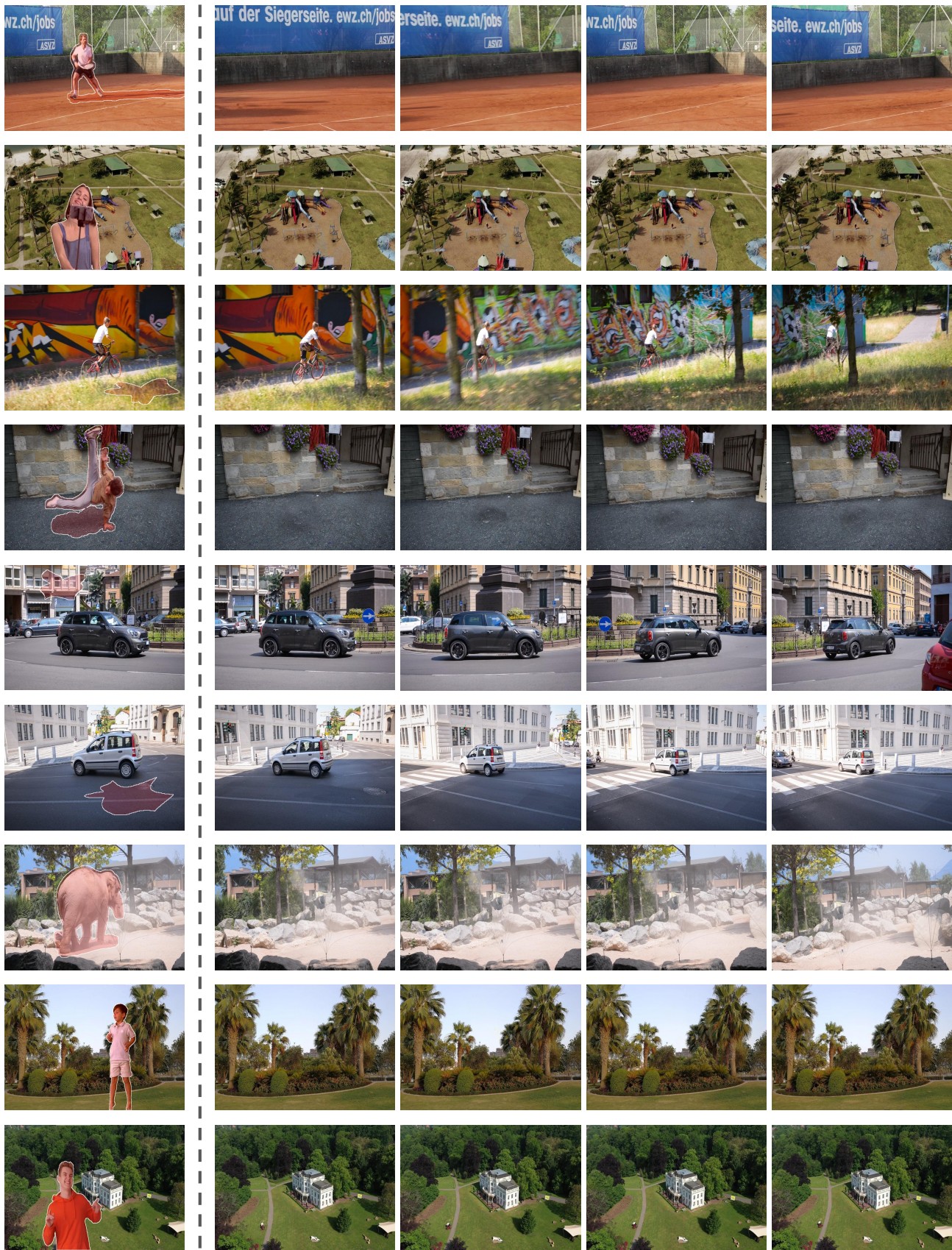

*Figure 9.* Additional qualitative results.

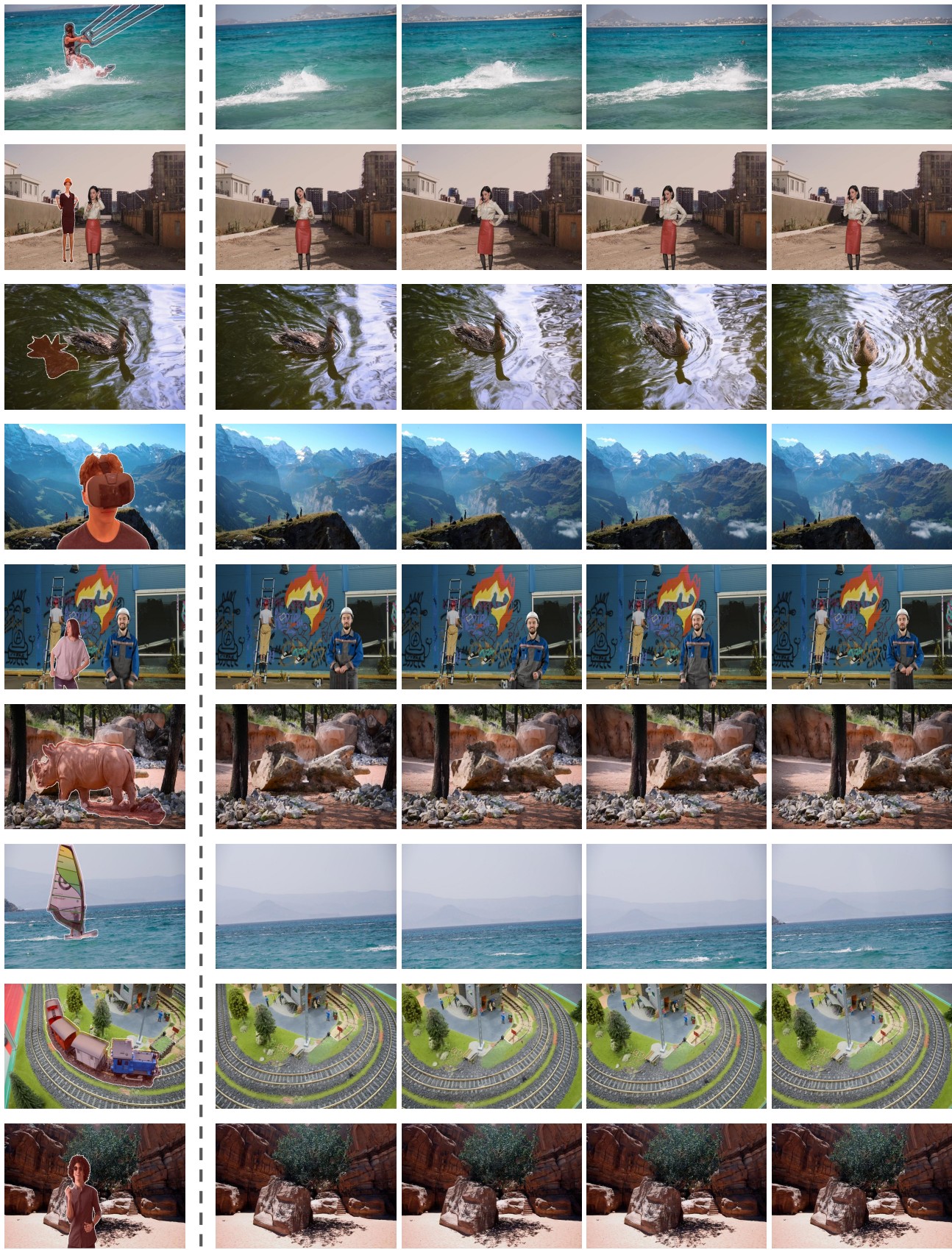

*Figure 10.* Additional qualitative results.

