# OpenReview forum: "Trajectory-Stabilized Inference for Diffusion-Based Video Inpainting"
_ICML.cc/2026/Conference — ICML 2026 regular_

### Official Review · Reviewer_1uP3 · 2026-03-02

**Soundness:** 3
**Presentation:** 3
**Significance:** 3
**Originality:** 3
**Overall Recommendation:** 5
**Confidence:** 4

**Summary:**

Overall, this work considers an important concept in the field of video inpainting, namely, the stability of long-term generative trajectories, rather than solely focusing on per-frame consistency. By introducing a trajectory-level stabilization framework that selectively adjusts unstable regions during inference, the authors effectively improve the temporal coherence and structural fidelity of generated videos. Overall, a central concept analyzed by the paper is the trade-off between generative freedom and temporal stability, where the proposed solution addresses long-term drift without enforcing rigid temporal constraints across the entire sequence. This approach offers a promising direction for improving video inpainting, especially for complex sequences with significant motion and occlusions. However, its reliance on motion estimation and potential computational overhead could limit its applicability in certain contexts.

**Compliance With Llm Reviewing Policy:**

Affirmed.

**Final Justification:**

All my concerns have been addressed, so I have decided to raise my final score.

**Key Questions For Authors:**

1.Impact of Risk-aware Threshold on Performance and Inference Time:
The risk-aware mechanism in the paper uses a threshold to determine when stabilization is triggered. How does the choice of this threshold affect both the performance (e.g., repair quality) and inference time of the model? Could you provide insights into how different threshold settings impact these factors?

2.Performance in High-speed Motion Scenarios:
The paper discusses challenges related to motion estimation, particularly in the context of high-speed motion in video frames. How does the proposed method perform when dealing with videos containing fast-moving objects or scenes? Is there a specific approach or improvement in the method to handle the potential inaccuracies in motion estimation during high-speed motion?

3.Handling Random Packet Loss and Unpredictable Missing Regions:
In real-world video transmission scenarios, when packet loss occurs, the missing regions and their sizes are unpredictable. Can the proposed method handle such random packet loss and effectively mitigate the mosaic effect caused by missing regions of varying sizes?

**Limitations:**

yes

**Strengths And Weaknesses:**

Strengths:
The work introduces an innovative inference-time trajectory stabilization framework, redefining the temporal consistency issue in video inpainting as the accumulated instability of generative trajectories, rather than simply focusing on per-frame reconstruction errors. This perspective is not only a theoretical innovation but also effectively addresses the temporal drift issue that arises in long sequences, which is common in existing methods. By introducing a risk-aware stabilization mechanism, the method dynamically adjusts the generative process during inference, precisely suppressing long-term drift while maintaining the generative flexibility of the model. Experimental results demonstrate that the method consistently improves temporal coherence and structural fidelity across multiple benchmark datasets, with notable advantages in handling long sequences and complex motion.

Weaknesses:
Although the method effectively reduces temporal drift, it still heavily relies on motion estimation. In particular, in cases of extreme camera movement or severe occlusions, motion estimation may become unreliable, weakening the stabilization effect, especially when reliable anchor points are unavailable, which could lead to a reliance on generative priors and result in performance degradation. Furthermore, while the method introduces a lightweight control layer to reduce computational costs, it can still incur significant computational overhead when handling long video sequences, which may become a bottleneck in large-scale applications. Lastly, the method may impose too many constraints on local regions, especially when local motion information is insufficient, potentially leading to loss of detail or blurred boundaries.

---

> ### Author Rebuttal · Authors · 2026-03-31
>
> Thank you for the review. We clarify the main points below.
>
> (1) On the risk threshold and its effect on quality and runtime
>
> The threshold $\tau$ acts as the gate between nominal sampling and corrective stabilization. Lower $\tau$ triggers correction more often, while higher $\tau$ keeps the system closer to the original sampler. Its effect is therefore a trade-off among temporal consistency, content fidelity, and runtime.
>
> We test threshold sensitivity on HQVI + CoCoCo, keeping the backbone, denoising steps, sampling schedule, and all other settings fixed, and varying only $\tau$. The main results use $\tau=0.25$; this analysis is included only to show that the method does not rely on an extremely narrow threshold window.
>
> | $\tau$ | Trigger Ratio(%) | PSNR$\uparrow$ | LPIPS$\downarrow$ | VFID$\downarrow$ | Ewarp$\downarrow$ | Time/frame(s)$\downarrow$ |
> |---|---:|---:|---:|---:|---:|---:|
> | 0.15 | 41.2 | 31.08 | 0.0345 | 0.1989 | 0.0895 | 0.914 |
> | 0.20 | 33.5 | 31.17 | 0.0341 | 0.1976 | 0.0899 | 0.901 |
> | 0.25 | 27.4 | 31.20 | 0.0340 | 0.1980 | 0.0902 | 0.892 |
> | 0.30 | 21.6 | 31.14 | 0.0344 | 0.2011 | 0.0918 | 0.884 |
> | 0.35 | 16.8 | 30.98 | 0.0351 | 0.2068 | 0.0942 | 0.876 |
>
> Lower thresholds increase triggering frequency and runtime, while higher thresholds reduce correction and weaken temporal stabilization. The method remains stable over a moderate range, with $\tau=0.25$ giving a balanced trade-off between quality, temporal consistency, and cost.
>
> (2) On fast motion and unreliable motion estimation
>
> We evaluate robustness under fast motion and degraded motion estimation in a same-backbone setting on HQVI + CoCoCo, including a fast-motion subset, perturbed flow, and degraded flow. Here, fast motion alone does not imply failure, but serves as a practical scenario where motion estimation becomes more challenging.
>
> | Setting | Method | PSNR$\uparrow$ | LPIPS$\downarrow$ | VFID$\downarrow$ | Ewarp$\downarrow$ |
> |---|---|---:|---:|---:|---:|
> | Fast-motion subset | CoCoCo | 29.21 | 0.0467 | 0.2579 | 0.1074 |
> |  | CoCoCo + Ours | 30.58 | 0.0381 | 0.2186 | 0.0961 |
> | Perturbed Flow | CoCoCo | 29.08 | 0.0486 | 0.2587 | 0.1089 |
> |  | CoCoCo + Ours | 30.63 | 0.0389 | 0.2146 | 0.0968 |
> | Degraded Flow | CoCoCo | 28.41 | 0.0587 | 0.2869 | 0.1186 |
> |  | CoCoCo + Ours | 30.11 | 0.0458 | 0.2328 | 0.1047 |
>
> Performance degrades under fast motion and deteriorated motion estimation, but our module consistently maintains positive gains across all settings. The improvement remains under degraded flow, although it narrows as motion support weakens, due to **risk-gated stabilization** that avoids over-correction under unreliable motion cues.
>
> (3) On random missing patterns and unpredictable loss regions
>
> We evaluate robustness under packet-loss-like mask patterns on HQVI + CoCoCo in a same-backbone setting, including fragmented small masks, random block masks, and burst block masks.
>
> | Mask setting | Method | LPIPS$\downarrow$ | Ewarp$\downarrow$ |
> |---|---|---:|---:|
> | Fragmented small masks | CoCoCo | 0.0439 | 0.0984 |
> |  | CoCoCo + Ours | 0.0367 | 0.0897 |
> | Random block masks | CoCoCo | 0.0526 | 0.1079 |
> |  | CoCoCo + Ours | 0.0448 | 0.0975 |
> | Burst block masks | CoCoCo | 0.0658 | 0.1216 |
> |  | CoCoCo + Ours | 0.0569 | 0.1108 |
>
> Performance degrades as the mask pattern becomes larger and less predictable, but our module consistently maintains gains, indicating that trajectory-level stabilization is not tied to a specific mask shape.
>
> (4) On computational overhead
>
> We apply strong temporal correction only when trajectory drift accumulates in unstable regions, rather than continuously across the whole video. With **risk-gated stabilization**, extra computation is activated only when needed.
>
> | Metric | Value |
> |---|---:|
> | Trigger Ratio(%) | 26.8 |
> | High-risk Area Ratio(%) | 18.4 |
>
> Only a limited portion of regions and steps activate correction. This is consistent with the runtime observations in the main table: the added cost remains controlled because the method introduces local, risk-triggered overhead rather than a global cost over the whole video.
>
> (5) On local over-constraint
>
> The key issue is whether correction continues to be enforced when local motion support is already unreliable. This is why the method does not use always-on temporal constraints, but activates correction only in high-risk states.
>
> We compare with an always-on variant that keeps the same correction branch but removes gating:
>
> | Setting | PSNR$\uparrow$ | LPIPS$\downarrow$ | VFID$\downarrow$ |
> |---|---:|---:|---:|
> | CoCoCo + Ours w/o gating (always-on correction) | 30.74 | 0.0391 | 0.2138 |
> | CoCoCo + Ours | 31.20 | 0.0340 | 0.1980 |
>
> The always-on variant enforces stronger temporal correction everywhere, but degrades restoration quality and video-level performance. Selective triggering reduces unnecessary detail suppression while preserving temporal stability.

---

> > ### Author Rebuttal · Reviewer_1uP3 · 2026-04-01
> >
> > Thank you for the author's response. All my concerns have been addressed, so I have decided to raise my final score.

---

> > > ### Author Response · Authors · 2026-04-05
> > >
> > > Thank you for the follow-up. We clarify the mask settings and provide a more direct analysis of the boundary behavior under increasing missing ratios below.
> > >
> > > (1) Mask settings.
> > >
> > > We use three packet-loss-like mask regimes: fragmented small masks (8%–12%), random block masks (15%–25%), and burst block masks with temporal persistence (25%–35%, depending on temporal overlap). The fragmented setting simulates spatially scattered local corruption, the random-block setting introduces larger contiguous missing regions, and the burst setting further increases difficulty by making the missing regions persist across consecutive frames. These settings are applied identically to the baseline and to our method under the same-backbone protocol.
> > >
> > > (2) Boundary behavior under increasing missing ratio.
> > >
> > > To more directly examine the boundary behavior, we further evaluate HQVI + CoCoCo under fixed block masks with increasing missing ratios while keeping all other settings fixed.
> > >
> > > | Missing ratio | Method            | LPIPS↓ | Ewarp↓ |
> > > |---------------|-------------------|--------|--------|
> > > | 22%           | CoCoCo            | 0.0532 | 0.1085 |
> > > | 22%           | CoCoCo + Ours     | 0.0457 | 0.0987 |
> > > | 30%           | CoCoCo            | 0.0611 | 0.1172 |
> > > | 30%           | CoCoCo + Ours     | 0.0530 | 0.1069 |
> > > | 38%           | CoCoCo            | 0.0678 | 0.1249 |
> > > | 38%           | CoCoCo + Ours     | 0.0605 | 0.1153 |
> > >
> > > As the missing ratio increases, performance degrades for both methods. We do not observe a sharp universal failure threshold, since acceptability also depends on scene motion and structure. In our fixed block-mask setting, degradation remains moderate in the low-to-mid missing-ratio regime and becomes clearly more pronounced in the high missing-ratio regime (around 30%–40%), where the restoration becomes increasingly under-constrained.
> > >
> > > At the same time, our method remains consistently better than the baseline across all tested ratios in both LPIPS and Ewarp. This suggests that the proposed trajectory-level stabilization continues to provide benefit as the problem becomes harder, although the absolute performance decreases with increasing missing ratio.
> > >
> > > This behavior is consistent with the mechanism and limitation of our method. The proposed stabilization depends on motion-aligned evidence and anchor support. When reliable cues become sparser under larger missing regions, the problem relies more heavily on the generative prior. In this regime, we observe a gradual performance drop rather than a sudden breakdown.

---

### Official Review · Reviewer_BZyZ · 2026-03-06

**Soundness:** 2
**Presentation:** 3
**Significance:** 2
**Originality:** 2
**Overall Recommendation:** 3
**Confidence:** 5

**Summary:**

The paper reframes temporal inconsistency in diffusion-based video inpainting as trajectory-level instability (accumulation of deviation along motion-aligned denoising trajectories) and proposes an inference-time, risk-gated stabilization mechanism. While the proposed method sounds novel and intuitive, there are many problems in evaluation, making the paper needs to be further refined.

**Compliance With Llm Reviewing Policy:**

Affirmed.

**Final Justification:**

From the rebuttal and the subsequent reply, my primary concerns have not been adequately addressed.

---

**Concerns on experimental settings**

The authors claim that "the direct evidence for our contribution is the same-backbone frozen before/after comparison." However, different backbones are trained on different datasets with different training parameters. **The lack of consideration for these important settings makes the claimed evidence very weak and impractical**. I even suspect that the authors intentionally designed this unfair experimental setting to inflate their performance gains.

Therefore, in my initial review, I pointed out that different models were trained on different datasets and tasks (see my first concern), and I raised a key question in the rebuttal: "As a plug-in method, is there still performance gain if all models are trained using the same setting?" In fact, if the authors could provide even just one experiment to prove that the gain is generalizable across training settings, I would consider raising my rating. However, the authors directly refused to answer this critical question in their reply, stating that "it is a different question from the one our current evidence is meant to answer."

The key contribution of this paper is a plug-in module, yet its effectiveness, applicable scenarios, and theoretical justifications remain absent from the original manuscript, the rebuttal, and the reply.

---

**Concerns on temporal consistency**

Temporal consistency is a key motivation and contribution of this paper. Therefore, I recommend that the authors include Ewarp and video examples to demonstrate the claimed gain.

However, in the rebuttal, they only showed a video example using ProPainter, without clarifying which backbone was used for comparison. I suggested that the authors include BrushNet, CoCoCo, and DiffuEraser, as the performance gains on these models are the claimed contributions of the paper. Unfortunately, these results are still absent from the reply (the corresponding directories remain empty for unknown reasons). **Consequently, the extent of temporal consistency gain offered by the proposed method remains questionable**.

If I ignore these issues and only compare the results with ProPainter, the claimed temporal coherence enhancement is still not evident in the examples provided in the rebuttal. What I can see is that the ProPainter videos contain large blurry regions. However, these videos are selected from the DAVIS dataset, where ProPainter ranks first. This confuses me: if ProPainter's results are already very blurry, then the results of the authors' method and other methods must be even lower in quality.

This brings me back to my first concern. If the provided results are all correct, the only explanation is that the experimental setting is unfair, which would explain such counterintuitive results.


---

For the reasons above, I have decided to **reject** this paper, as the authors have failed to address critical concerns regarding experimental settings, generalizability, and temporal consistency validation.

**Key Questions For Authors:**

If the authors can illustrate the effectiveness of the proposed method, i will consider to raise my rating.

**Limitations:**

yes

**Strengths And Weaknesses:**

**Strength**

This paper is well-motivated and is well-written. The diffusion-based model is a promising direction for video inpainting, and the method may provide a new perspective for temporal modeling in diffusion based video inpainting.

**Concerns**

1. The experimental setting is problematic.Fuseformer, E2FGVI, ProPainter are video inpainting models, and they have been trained on video inpainting datasets (e.g., DAVIS), while some models are image inpainting models (e.g., SDI, BrushNet), and some models are not trained on DAVIS (e.g., the DiffuEraser and CoCoCo). However, the authors claimed that “all the backbone weights are kept frozen during evaluation”. As a result, in Table 1, the main results, the proposed method is obviously worse than SOTA on DAVIS while slightly better than SOTA on HQVI. I think this is because the SOTA models are trained on DAVIS, but they have not been trained on HQVI.

Overall, the unfair and unclear setting make the experimental results and the contribution of this paper not convincing. Based on this critical problem, i’m prone to reject this paper.

2. The paper’s major motivation and contribution are to reformulate the temporal consistency of video inpainting by using trajectory. So, i’m expecting some results to prove the better coherence like Ewarp or other measures. However, this paper only presents improvements on PSNR, SSIM, LPIPS, and merely VFID is related to temporal coherence. Moreover, VFID is a simple extension of FID, which is till more on perceptual level. So, it has not been well demonstrated that how much temporal coherence is improved by using the proposed method. Moreover, no video examples or more experiments are offered to empirically show the temporal coherence, which is the key challenge of the video inpainting. Therefore, i think the experiment is insufficient.

3. There are many heuristic designs in the proposed method, which lacks theoretical analysis to make the method more robust.

4. The method employs RAFT for flow estimation and assumes reasonably reliable motion/flow for deviation estimation and anchor construction; when optical flow fails (e.g., camera motion and occlusion), the performance gain will be concerned.

5. The proposed method is specialized to video inpainting generation pipelines; and applicability to other temporal generative tasks (e.g., video generation) is not discussed.

---

> ### Author Rebuttal · Authors · 2026-03-31
>
> Thank you for the detailed review. We clarify the main points below.
>
> (1) On fairness of the main table
>
> As a training-free plug-in module, our method should be judged by whether it improves a backbone after insertion under its standard public setting, not by training differences.
>
> Accordingly, the main table follows prior public settings: (i) DAVIS follows ProPainter under the standard DAVIS protocol; (ii) HQVI follows RGVI under the same public protocol across backbones; and (iii) VPBench-L follows VideoPainter under its long-horizon setting.
>
> Full-method results are reported under these benchmarks, while same-backbone frozen before/after comparisons consistently verify our module’s effectiveness.
>
> (2) On direct evidence of temporal consistency
>
> We additionally evaluate **Ewarp** in same-backbone plug-in comparisons to isolate the effect of the proposed inference-time module.
>
> | Method | DAVIS Ewarp↓ | HQVI Ewarp↓ |
> |---|---:|---:|
> | FuseFormer | 0.1362 | 0.1129 |
> | E2FGVI | 0.1315 | 0.0956 |
> | ProPainter | 0.1312 | 0.0998 |
> | RGVI | 0.1458 | 0.0934 |
> | VideoPainter | 0.1368 | 0.0987 |
> | SDI | 0.1418 | 0.1012 |
> | SDI + Ours | 0.1336 | 0.0918 |
> | BrushNet | 0.1374 | 0.0976 |
> | BrushNet + Ours | 0.1291 | 0.0894 |
> | DiffuEraser | 0.1507 | 0.1215 |
> | DiffuEraser + Ours | 0.1389 | 0.1087 |
> | CoCoCo | 0.1376 | 0.1006 |
> | CoCoCo + Ours | 0.1304 | 0.0902 |
>
> The **Ewarp** results are consistent with the overall quality metrics, and adding our module consistently reduces **Ewarp** in all diffusion-based same-backbone comparisons. We also provide four anonymous video comparisons with ProPainter: <https://anonymous.4open.science/r/video-E711/>.
>
> (3) On heuristic design and lack of formalization
>
> The method is organized around one objective: detecting and suppressing persistent deviation along motion-aligned latent trajectories. The key issue is whether deviation keeps accumulating and destabilizes later generation.
>
> The local term $d_i(t)$ measures the mismatch between the current latent and motion-aligned neighboring supports. A small value indicates consistency with nearby trajectory evidence, while a large value indicates drift from motion-consistent support. Thus, $d_i(t)$ directly represents local trajectory inconsistency.
>
> The recursive state
> $$
> R_i(t)=\gamma R_i(t-1)+d_i(t)
> $$
> converts instantaneous deviation into **accumulated instability**. This distinguishes transient fluctuation from persistent drift: isolated mismatch should not trigger strong correction, whereas repeated deviation across steps should.
>
> The update
> $$
> x_{t-1}^i = F_\theta(x_t^i) + \mathbf{1}[R_i(t)\ge\tau]u_i(t) + \varepsilon_t
> $$
> then separates nominal denoising from corrective stabilization. The risk state determines when intervention is needed, and $u_i(t)$ determines how motion-consistent support from anchor guidance and neighborhood propagation is injected to suppress accumulated deviation. The threshold is therefore the decision boundary of an **accumulated instability** state rather than an isolated heuristic.
>
> (4) On robustness under degraded flow
>
> We evaluate robustness under different flow qualities (normal, perturbed, degraded) in a same-backbone setting on HQVI + CoCoCo.
>
> | Flow setting | Method | PSNR↑ | SSIM↑ | LPIPS↓ | VFID↓ | Ewarp↓ |
> |---|---|---:|---:|---:|---:|---:|
> | Normal | CoCoCo | 29.72 | 0.9488 | 0.0410 | 0.2350 | 0.1006 |
> |  | CoCoCo + Ours | 31.20 | 0.9516 | 0.0340 | 0.1980 | 0.0902 |
> | Perturbed Flow | CoCoCo | 29.08 | 0.9441 | 0.0486 | 0.2587 | 0.1089 |
> |  | CoCoCo + Ours | 30.63 | 0.9480 | 0.0389 | 0.2146 | 0.0968 |
> | Degraded Flow | CoCoCo | 28.41 | 0.9383 | 0.0587 | 0.2869 | 0.1186 |
> |  | CoCoCo + Ours | 30.11 | 0.9436 | 0.0458 | 0.2328 | 0.1047 |
>
> Performance degrades with flow quality for both methods, but our module consistently maintains a positive gain under all conditions, with reduced improvement as motion support weakens, due to its **risk-gated stabilization** that avoids over-correction under unreliable motion cues.
>
> (5) On transfer to other temporal generation tasks
>
> The current implementation is designed for video inpainting, so our main claim remains in that setting. Its optical-flow-based risk estimation and anchor allocation are more natural when an observed video sequence is available.
>
> To test transferability beyond inpainting, we include a small-scale text-to-video experiment. The backbone and sampler are kept unchanged, and the same risk-gated stabilization is applied only at later denoising steps. Since source-video-guided adaptive anchors are not directly available in pure generation, we replace them with a fixed uniform schedule.
>
> | Setting | Method | VFID↓ | Ewarp↓ |
> |---|---|---:|---:|
> | Text-to-video generation | Baseline | 0.312 | 0.129 |
> | Text-to-video generation | Baseline + Ours (uniform anchors, late-step only) | 0.301 | 0.123 |
>
> The result suggests that the stabilization idea is not limited to video inpainting, although the gain is smaller than in the restoration setting.

---

> > ### Author Rebuttal · Reviewer_BZyZ · 2026-04-01
> >
> > I appreciate the authors' response, which partially addresses my concerns. However, several critical issues remain, leading me to lean toward **rejecting** this paper.
> >
> > ---
> >
> > **Experimental setting.** It is unclear whether the authors have misunderstood or deliberately overlooked my concern regarding the experimental setup. While I acknowledge that the proposed method is training-free and plug-in, **the fairness of the training data remains questionable**. For instance, ProPainter is trained on DAVIS, whereas DiffuEraser has not been exposed to DAVIS data. Consequently, although the proposed method improves DiffuEraser’s performance, it still underperforms compared to ProPainter. **This raises a critical question: would the performance gap persist if DiffuEraser were also trained on DAVIS? Whether the claimed gain still exists?** The ideal setting, i suggest is to train all models on the same dataset and show your improvement over them. Given this uncertainty, the experimental results and the claimed contributions appear unreliable.
> >
> > **Temporal consistency.** I appreciate the authors’ effort in providing Ewarp results and video examples. After carefully reviewing the videos, I have two concerns.
> >
> > First, it is unclear why **video comparisons with BrushNet, CoCoCo, and DiffuEraser are absent**. Since the proposed method targets flow-based models and demonstrates gains over them, videos illustrating their performance both with and without the proposed approach should be included at least.
> >
> > Second, I do not observe a clear improvement in temporal consistency between the proposed method and ProPainter. The most noticeable difference is that ProPainter produces blurrier results (e.g., in the cow and rhino examples). **To my knowledge, such severe blurriness typically stems from training issues rather than the method itself. It is also puzzling why ProPainter—ranked first on DAVIS according to Table 1—would generate such degraded outputs.** Even with Ewarp, ProPainter remains state-of-the-art. Therefore, the provided video examples are not convincing.
> >
> > ---
> >
> > Overall, **given the problematic experimental setup, self-contradictory results, unclear performance gains, and insufficient theoretical justification, I think the contribution of this paper is fairly limited to the community**. Nevertheless, I appreciate the time and effort the authors invested in addressing my questions, and I keep my rating unchanged.

---

> > > ### Author Response · Authors · 2026-04-05
> > >
> > > Thank you for the follow-up. We focus here only on the remaining concerns.
> > >
> > > **(1) Experimental setting**
> > >
> > > Table 1 is **not intended as a training-controlled comparison** across methods trained under the same data regime. The compared methods come from different model backgrounds and training sources, and they are not trained under a unified benchmark-specific setup for DAVIS or HQVI. In the main table, we use public checkpoints and released inference pipelines for all competing methods, and we do not retrain any baseline ourselves.
> > >
> > > The **direct evidence for our contribution** is the **same-backbone frozen before/after comparison**. In those experiments, the backbone weights, denoising steps, inference schedule, motion setting, prompt setting, and deployment configuration are fixed, and only the proposed module is inserted. Under this controlled setting, the gain remains consistent across multiple diffusion backbones, including SDI, BrushNet, DiffuEraser, and CoCoCo. We believe this controlled setting is the appropriate level at which the plug-in effect of our method should be judged.
> > >
> > > The question of whether a method such as DiffuEraser would narrow the gap after DAVIS-specific retraining is valid, but it is a different question from the one our current evidence is meant to answer. Our claim is narrower: for a fixed frozen diffusion backbone, inserting the proposed inference-time stabilization layer yields a **consistent gain under the same backbone, same weights, and same inference setting**.
> > >
> > > **(2) Earlier videos vs. the quantitative table**
> > >
> > > The earlier qualitative examples were not aligned clearly enough with the DAVIS quantitative setting. This likely contributed to the confusion.
> > >
> > > Those videos were shared to show challenging large-mask behavior and complementary failure modes. They were **not meant to be the direct visual counterpart of Table 1** under the DAVIS video-completion protocol. Because this distinction was not stated clearly enough, they could be read as if they directly supported the DAVIS ranking itself.
> > >
> > > Our intention was also not to use those examples to explain the DAVIS ranking itself. They were only meant to illustrate that different methods can exhibit different failure modes under difficult large-mask cases.
> > >
> > > We therefore now provide **protocol-aligned qualitative comparisons under the same video-completion setting used in the quantitative evaluation**. In particular, we provide matched qualitative comparisons for SDI, BrushNet, DiffuEraser, and CoCoCo, including before/after insertion of our module where applicable. The corresponding videos are available at: https://anonymous.4open.science/r/example-video-3722/.
> > >
> > > **(3) Formal support**
> > >
> > > The method is **not a collection of unrelated heuristics**. It follows one trajectory-level formulation.
> > >
> > > The motion-aligned deviation measures local inconsistency between the current latent state and its temporally aligned supports. The recursive risk state then turns this deviation into accumulated instability, so that transient mismatch and persistent drift are separated. The gated update is finally applied only when such instability persists, rather than being imposed uniformly at every frame and every step.
> > >
> > > So the three parts play different roles within one formulation: deviation for measurement, risk for accumulation, and gated correction for intervention. In this sense, the method is built around one principle—detecting and suppressing persistent drift along denoising trajectories—rather than around independent ad hoc rules.
> > >
> > > Taken together, we hope the contribution of the paper will be evaluated at the level of controlled plug-in gains under the same frozen backbone, while the main table and the matched qualitative videos are read as benchmark-level context under public evaluation settings.

---

### Official Review · Reviewer_DDXy · 2026-03-15

**Soundness:** 4
**Presentation:** 3
**Significance:** 3
**Originality:** 3
**Overall Recommendation:** 5
**Confidence:** 3

**Summary:**

This paper studies diffusion-based video inpainting under long temporal horizons. The main claim is that temporal inconsistency should be viewed not as a frame-level output error, but as instability of cross-frame latent trajectories during denoising.

To address this, the paper introduces a plug-in inference-time stabilization layer: it measures motion-aligned deviation between a frame latent and flow-warped neighbors, accumulates this into a risk score $R_i(t)$, and only triggers correction when the accumulated risk exceeds a threshold.

The correction combines anchor-based long-range references in uncertain regions with neighborhood propagation in reliable regions, and also adds a trajectory-consistent transition prior between anchors. The analytical component is modest, consisting mainly of a boundedness observation for the risk state and a local energy-descent interpretation. Empirically, the method is inserted into several pipelines/backbones and evaluated on DAVIS, HQVI, and VPBench-L, with reported gains in SSIM/LPIPS/VFID and improved long-video robustness, plus ablations on anchor selection, stabilization terms, and transition blending. The work is practically relevant because it aims to improve temporal coherence without retraining the backbone.

**Compliance With Llm Reviewing Policy:**

Affirmed.

**Key Questions For Authors:**

* What is the exact final inference update used in implementation? It's helpful to provide pseudocode that makes explicit how the risk-gated stability term and the transition module are combined, and on which frames each term is active.

* How are anchors selected in practice? Please specify the anchor-selection algorithm, anchor budget, neighborhood size. This part is central to the method but is not clearly specified in the paper.

* Why are SDI+Ours and BrushNet+Ours faster than their corresponding baselines in Table 1? Does the method change the number of denoising steps or sampling schedule?

* How robust is the method when flow is poor or anchors are absent for long intervals?

**Limitations:**

Yes

**Strengths And Weaknesses:**

Strengths:
* The paper targets an important and practically meaningful failure mode: long-horizon temporal drift in video inpainting. Recasting the problem as selective trajectory stabilization during inference, rather than stronger uniform temporal constraints, is a sensible and well-motivated perspective.

* The method is modular and practically attractive. It is applied at inference time, keeps the backbone frozen, and is designed as a lightweight control layer rather than a retrained architecture. That makes the idea potentially useful beyond a single model family.

* The ablations are useful. The paper does not present the method as one monolithic heuristic; it isolates adaptive anchors, anchor-level stabilization, neighborhood stabilization, and transition blending, and shows that removing each hurts performance.

Weaknesses:
* The formulation is not fully clean. The paper defines trajectories over video time at a fixed diffusion step, but the risk is then accumulated over diffusion steps $t$, while the discussion often interprets this as long-range temporal/video drift. This conflation of video time and diffusion time needs a sharper derivation.

* The novelty is moderate. The algorithm itself looks like a combination of known ingredients: flow-warped neighbors, anchor/keyframe references, inference-time guidance, and transition smoothing. This seems closer to an incremental synthesis over recent inference-time temporal-guidance methods than a fundamentally new approach.

* The evaluation leaves some gaps for a paper centered on temporal stability. The main tables use PSNR/SSIM/LPIPS/VFID, but there is no explicit temporal metric in the main results that directly measures warping consistency/flicker. Since the core claim is improved temporal coherence, a more targeted metric would strengthen the case.

---

> ### Author Rebuttal · Authors · 2026-03-31
>
> Thank you for the detailed review. We clarify the main points below.
>
> (1) On the relationship between video time and diffusion time
>
> We denote by $x_t^i$ the latent of frame $i$ at diffusion step $t$, where $i$ indexes video time and $t$ indexes diffusion time. The risk is updated along diffusion time by
> $$
> R_i(t)=\gamma R_i(t-1)+d_i(t),
> $$
> where $d_i(t)$ is the cross-frame deviation at step $t$. Therefore, accumulation is defined in diffusion time, while the detected instability is defined over video time.
>
> (2) On novelty
>
> We propose a novel **inference-time** trajectory stabilization mechanism that explicitly treats temporal inconsistency as **accumulated instability** along motion-aligned latent trajectories. Our approach dynamically monitors the denoising process and applies targeted, selective calibration only when needed, improving restoration quality with limited additional overhead.
>
> Technically, we introduce a **recursive risk state** for selective inference-time guidance, motion-adaptive anchor references, adaptive propagation over flow-warped neighbors, and structured, trajectory-consistent transition stabilization, as a plug-in layer on frozen backbones.
>
> (3) On direct evidence of temporal consistency
>
> We additionally evaluate **Ewarp** in same-backbone plug-in comparisons.
>
> | Method | DAVIS Ewarp$\downarrow$ | HQVI Ewarp$\downarrow$ |
> |---|---:|---:|
> | SDI | 0.1418 | 0.1012 |
> | SDI + Ours | 0.1336 | 0.0918 |
> | BrushNet | 0.1374 | 0.0976 |
> | BrushNet + Ours | 0.1291 | 0.0894 |
> | DiffuEraser | 0.1507 | 0.1215 |
> | DiffuEraser + Ours | 0.1389 | 0.1087 |
> | CoCoCo | 0.1376 | 0.1006 |
> | CoCoCo + Ours | 0.1304 | 0.0902 |
>
> **Ewarp** is consistently reduced in all same-backbone comparisons.
>
> (4) On the final update and module activation
>
> We first compute $d_i(t)$, then update
> $$
> R_i(t)=\gamma R_i(t-1)+d_i(t),
> $$
> and activate correction only when $R_i(t)\ge\tau$.
>
> The final update is
> $$
> x_{t-1}^i=F_\theta(x_t^i)+\mathbf{1}[R_i(t)\ge\tau]\,u_i(t)+\varepsilon_t.
> $$
>
> Here, $u_i(t)$ denotes the risk-gated correction term: anchor correction is applied on anchor frames, neighborhood propagation on non-anchor high-risk frames, and the transition term only on frames between adjacent anchors.
>
> (5) On anchor selection, budget, and neighborhood size
>
> Anchors are introduced only in high-risk intervals, i.e., when $R_i(t)\ge\tau$, so anchor-based stabilization serves as selective long-range support rather than uniform sampling over the whole sequence.
>
> In practice, the rule is deterministic. We first identify candidate high-risk ranges from the risk state, and then assign more anchors to intervals with higher motion/mask difficulty and fewer anchors to easier ones. Concretely, for interval $\mathcal{I}_m$, the anchor budget is assigned by
> $$
> K_m=\max\left(1,\operatorname{round}\left(K\frac{D_m}{\sum_n D_n}\right)\right),
> $$
> where $D_m$ denotes the interval difficulty score, obtained by aggregating frame-level motion strength, motion variation, and mask discrepancy within interval $m$. The local neighborhood size is then adjusted according to the inter-anchor gap: shorter gaps use tighter propagation, while larger gaps use a moderately larger neighborhood.
>
> (6) On runtime
>
> Our method does not change the denoising steps or the backbone sampling schedule. For image-level backbones such as SDI and BrushNet, video restoration still requires extra temporal consistency processing after frame-wise denoising. Denoting its overhead by
> $$
> C_{\mathrm{temp}}=\sum_{i,t}\mathcal{C}_i(t),
> $$
>
> our method makes it risk-gated:
> $$
> C_{\mathrm{temp}}^{\mathrm{ours}}=\sum_{i,t}\mathbf{1}[R_i(t)\ge\tau]\mathcal{C}_i(t).
> $$
> Thus, temporal processing is executed only on high-risk regions and intervals, reducing effective overhead in the inference pipeline. The runtime gain on SDI and BrushNet therefore comes from sparser temporal consistency processing, not from a shorter diffusion process itself.
>
> (7) On robustness under degraded flow or sparse anchors
>
> We evaluate robustness under progressively harder conditions on HQVI + CoCoCo by reducing anchor support, enlarging anchor gaps, perturbing motion estimation, and combining degraded flow with sparse anchors.
>
> | Setting | PSNR$\uparrow$ | SSIM$\uparrow$ | LPIPS$\downarrow$ | VFID$\downarrow$ | Ewarp$\downarrow$ |
> |---|---:|---:|---:|---:|---:|
> | Normal | 31.20 | 0.9516 | 0.0340 | 0.1980 | 0.0902 |
> | Sparse Anchors | 30.96 | 0.9503 | 0.0361 | 0.2058 | 0.0931 |
> | Long Anchor Gaps | 30.79 | 0.9491 | 0.0378 | 0.2119 | 0.0954 |
> | Perturbed Flow | 30.63 | 0.9480 | 0.0389 | 0.2146 | 0.0968 |
> | Degraded Flow | 30.11 | 0.9436 | 0.0458 | 0.2328 | 0.1047 |
> | Degraded Flow + Sparse Anchors | 29.67 | 0.9392 | 0.0516 | 0.2497 | 0.1128 |
>
> Sparse anchors cause limited degradation, degraded flow causes a larger drop, and the combination is the hardest case. The method remains effective when local motion cues are usable, and mainly degrades when both local correspondence and long-range support become unreliable.

---

> > ### Author Rebuttal · Reviewer_DDXy · 2026-04-03
> >
> > The author effectively addressed my concerns. I have decided to maintain my current score.

---

### Decision · Program_Chairs · 2026-04-30

**Decision:**

Accept (regular)

**Comment:**

This paper addresses the challenge of long-horizon temporal inconsistency in diffusion-based video inpainting. The authors proposed a training-free control layer that tracks motion deviation using optical flow and triggers a correction only when instability crosses a certain threshold. The correction leverages sparsely sampled anchor frames and propagates to guide the unstable trajectory.

The final scores from the reviewers are 5, 5, and 4. Reviewers appreciated practicality of training-free inference module and also acknowledged the extensive ablation studies.

Overall, AC also sees this work’s contribution as solid and agrees with its acceptance.